# FMRP expression in primary breast tumor cells correlates with recurrence and specific site of metastasis

E. Caredda[1,2], G. Pedini[1], F. D'Amico[3,4], M. G. Scioli[3], L. Pacini[1,5], P. Orsaria[6], G. Vanni[7], O. C. Buonomo[7], A. Orlandi[3], C. Bagni[1,8], L. Palombi [1]*

1 Department of Biomedicine and Prevention, Faculty of Medicine, University of Rome Tor Vergata, Rome, Italy, 2 Directorate-General for Health Prevention, Ministry of Health, Rome, Italy, 3 Anatomic Pathology, Department Biomedicine and Prevention, Faculty of Medicine, Tor Vergata University Hospital, Rome, Italy, 4 Infectious Diseases Unit, Niguarda Hospital, Milan, Italy, 5 UniCamillus, Saint Camillus International, Faculty of Medicine, University of Health and Medical Sciences, Rome, Italy, 6 Department of Breast Surgery, University Campus Bio-Medico, Rome, Italy, 7 Department of Surgery, Faculty of Medicine, Tor Vergata University Hospital, Rome, Italy, 8 Department of Fundamental Neurosciences (DNF), Faculty of Biology and Medicine, University of Lausanne, Lausanne, Switzerland

* leonardo.palombi@gmail.com

**Data Availability Statement:** All data are available with the Harvard Dataverse. Below the final DOI of the submitted data repository entry for Harvard Dataverse: https://dataverse.harvard.edu/dataset.

## Abstract

Breast cancer is the most common cancer among women worldwide. Molecular and clinical evidence indicated that Fragile X Messenger Ribonucleoprotein 1 (FMRP) plays a role in different types of cancer, including breast cancer. FMRP is an RNA binding protein that regulates the metabolism of a large group of mRNAs coding for proteins involved in both neural processes and in epithelial-mesenchymal transition, a pivotal mechanism that in cancer is associated to tumor progression, aggressiveness and chemoresistance. Here, we carried out a retrospective case-control study of 127 patients, to study the expression of FMRP and its correlation with metastasis formation in breast cancer. Consistent with previous findings, we found that FMRP levels are high in tumor tissue. Two categories have been analyzed, tumor with no metastases (referred as control tumors, 84 patients) and tumor with distant metastatic repetition, (referred as cases, 43 patients), with a follow-up of 7 years (mean). We found that FMRP levels were lower in both the nuclei and the cytoplasm in the cases compared to control tumors. Next, within the category cases (tumor with metastases) we evaluated FMRP expression in the specific sites of metastasis revealing a nuclear staining of FMRP. In addition, FMRP expression in both the nuclear and cytoplasmic compartment was significantly lower in patients who developed brain and bone metastases and higher in hepatic and pulmonary sites. While further studies are required to explore the underlying molecular mechanisms of FMRP expression and direct or inverse correlation with the secondary metastatic site, our findings suggest that FMRP levels might be considered a prognostic factor for site-specific metastasis.

xhtml?persistentId=doi:10.7910/DVN/BGKJC0 To avoid interrupted downloads, time outs or other failures you can use the Downloading via URL (Use the Download URL in a Wget command or a download manager) https://dataverse.harvard.edu/api/access/datafile/7068840.

**Funding:** This work was supported by Telethon GGP20137 and PRIN 201789LFKB MIUR, to CB. The funders had no role in study design, data collection and analysis, decision to publish, or preparation of the manuscript. Other Authors received no specific funding for this work.

**Competing interests:** The authors have declared that no competing interests exist.

## Introduction

Breast cancer (BC) is the most common cancer among women worldwide. In 2020, the World Health Organization (WHO) estimated that 2,261,419 of the 10 million cancer cases were new BC cases (https://www.who.int/news-room/fact-sheets/detail/cancer), and despite advances in the diagnosis, treatment, and prevention of the disease, it was the leading cause of cancer mortality among women with 685,000 (6.8%) deaths worldwide (https://www.who.int/news-room/fact-sheets/detail/cancer). The poor prognosis of BC is mainly due to the development of distant metastases [1–3]. Therefore, the identification of prognostic markers of site-specific metastasis might contribute to develop personalized therapies and increase patient survival.

In recent years, several molecular subtypes of breast cancer have been identified, classified according to differences in gene expression, clinical features, and response to therapy [4]. Their definition is mainly based on receptor pattern, generally through IHC analysis, evaluating estrogen receptor (ER), progesterone receptor (PR), human epidermal growth factor receptor 2 (HER2), and cell proliferation marker ki67 factor [5]. Therefore, we can recognize five main molecular subtypes: the Luminal A, B negative and B positive (expressing HER2) and the Non-Luminal such as HER2 positive and Triple negative breast cancer (TNBC) or Basal-like. Based on this immune-histochemical panel, validated according to a gene expression profile [6, 7], the different pathological behaviors are described; this panel allows clinicians to consider distinct biological features before selecting appropriate therapeutic strategies [8, 9]. The five main molecular subtypes of BC are different in terms of the primary tumor characteristics, aggressiveness, response to chemotherapy [10], and their ability to metastasize to distant organs [11]. For example, the TNBC metastasizes to the lymph less than other classes of BC and maintains a limited growth (volume) [12, 13]; this subtype is characterized by an aggressive clinical course and, as also HER2, has high distant recurrence rates that decrease after the first few years (5-year survival). Luminal subtypes of BC spread more frequently by the lymphatic way, have lower rate of metastasis maintained over several years.

Metastasis occurs when a few tumor cells leave the primary tumor, enter into the lymphatic or blood system, and reach distant tissues or parenchyma, forming a colony in a secondary organ. Epithelial-mesenchymal transition (EMT) is an important mechanism associated with this process. EMT allows epithelial cells to acquire mesenchymal features as the ability to migrate and invade distant tissues, and it is a crucial event in tumor progression and metastasis [14–16]. At the molecular level, this process is characterized by the down-regulation of epithelial markers and the up-regulation of mesenchymal markers and matrix metalloproteinase enzymes, which are able to digest the scaffold of the extra-cellular matrix [17]. Due to these findings, an aberrant form of EMT has been implicated as a trigger for metastasis and is a potential target of anticancer therapy [18].

The Fragile X Messenger Ribonucleoprotein 1 (FMRP), absent or dysfunctional in the Fragile X Syndrome (FXS), it is an RNA binding protein whose activity is essential for brain functions [19, 20]. In the past decade growing evidence indicated that FMRP is involved in different molecular mechanisms associated with cancer onset, progression and metastasis [21, 22]. A role of FMRP has been shown in two brain tumors, the astrocytoma [23] and the glioblastoma [24]. In this last case, FMRP levels correlate with tumor proliferation and overall patient survival. Moreover, FMRP regulates the necroptotic pathway in colon cancer [25], the invasive behavior of cancer cells in melanoma [26] and in intrahepatic cholangiocarcinoma [27]. In the context of BC FMRP binds mRNAs regulating EMT [28, 29]. As a whole a series of consistent evidence support a key role of FMRP in different step of tumor progression.

In breast cancer, the analysis of four independent breast cancer datasets revealed that the overexpression of *FMR1* mRNA, which codes for FMRP, is associated with an increased risk of

lung metastasis formation, a correlation independent from the expression of estrogen receptor. Specifically, *FMR1* mRNA expression is increased in the most aggressive molecular subtype (TNBC or Basal-like) compared to the ER/PgR-positive and HER2-positive subtypes [29]. The most aggressive Basal-like subtype is strongly associated with pulmonary metastasis [30, 31], regardless of the involvement of the lymph nodes at diagnosis [32, 33].

In this study, we investigated the expression of FMRP in different molecular subtypes of BC and relate it to the outcome of metastasis. Our findings, show that FMRP is expressed in both the nucleus and cytoplasm of breast cancer cells and that different levels of FMRP correlate with specific metastatic sites suggesting a possible role as prognostic factor for site-specific metastasis.

## Materials and methods

### Patient ethical approval

The study was approved by the institutional review board at the University Hospital of Tor Vergata and conducted according to the current ethical guidelines. Written informed consent for the participation in retrospective studies was obtained from all patients at the time the surgery was performed, as defined by the protocols of the Independent Ethics Committee of the University Hospital of Tor Vergata.

### Study design and patient enrollment

This work is a retrospective study of 127 patients, who underwent surgery for primary BC from January 2000 to July 2013 at the University Hospital of Tor Vergata (Rome, Italy). Patients were enrolled according to defined eligibility criteria: same length of observation period between cases and controls, and diagnosis, surgery, radiochemotherapy, and follow-up conducted at the University Hospital of Tor Vergata (Rome, Italy). It is based on the correlation of FMRP expression with the anatomo-pathological data of the patients. Two groups of patients were analyzed: a group with tumor and distant metastatic (named cases) and a group with tumor and lack of metastases (named controls), paired at 1:2 ratio by age, age of onset, and duration of observation (mean 7 years). Cases and controls, previously described in Buonomo et al [11], were selected randomly and blind by the operator performing the cutting of the paraffin-embedded sample. The analysis of FMRP detection was carried out by the anatomo-pathologist blind to the cases or controls. A database was created were the score of FMRP staining and the clinical database using anonymized histological numbers. The risk of metastasis in controls was assumed to be 0.06 (6% of patients with BC developed one or more metastases). The alpha error and power of the study were fixed at 5% and 80%, respectively. The BC subtypes were identified according to the clinicopathological criteria recommended by the St. Gallen International Expert Consensus Report 2013 [10].

### Scoring system

All human tissues were collected following standardized procedures, including obtaining informed consent. The histopathological diagnosis of the tumors was based according to the WHO International Classification of Disease for Oncology. The clinical staging was determined by the TNM Staging System and the Elston and Ellis grading System. The malignancy of infiltrating carcinomas was scored according to the Scarff-Bloom-Richardson classification. Each sample was histopathologically evaluated to ensure the presence of at least 80% of tumor cells. The medical records of all the patients were examined to obtain clinical and histopathological information.

Patients were categorized based on the receptor status of their primary tumor as follows: luminal A (ER+ or PR+, and HER2-); luminal B HER2- (ER+, HER2-, and at least one of Ki-67 "high" or PR "negative or low"); luminal B HER2+ (ER+, HER2-overexpressed or amplified, any Ki-67, any PR); HER2 (ER- or PR-, and HER2+), and basal (ER- or PR-, and HER2-). ER and PR status were determined using immunohistochemistry (IHC). Tumors were considered HER2-positive only if they either had 3+ IHC staining score (strong, complete membrane-staining in > 30% of cancer cells) or showed HER2 amplification (ratio > 2) with fluorescence in situ hybridization (FISH). In the absence of positive FISH data, tumors that had 2+ IHC score were considered negative for HER2. Tumors were also classified as luminal or non-luminal according to the expression of hormone receptors (ER and PR).

## Immunohistochemistry

For IHC, sections were placed in 10% buffered formalin for 24 h, dehydrated, and embedded in paraffin. Serial 4-μm thick sections were deparaffinized, rehydrated, processed with EnVision, Flex+ kit (Dako), and incubated with rabbit polyclonal anti-α-FMRP (1:250) rAM2 [34]. Positive and negative controls were included. Cytoplasmic semi-quantitative evaluation of α-FMRP immunostaining was performed using a grading system in arbitrary units as follows: absence of positivity (0), weak (1), moderate (2), and strong (3) [35]. FMRP expression in the nucleus was evaluated with a semi-quantitative method and expressed as percentage of positive cells versus total; FMRP expression in the nucleus and cytoplasm was evaluated by two independent researchers using the Eclipse E600 microscope (Nikon, Japan), and images were acquired by ACT-1 software (Nikon) at 200× magnification. The inter-observer reproducibility was > 95%. For each case, the ratio of the score to the number of fields analyzed was determined. The slices in the study were chosen from non-diagnostic material.

## Statistical analysis

The results were collected in a database, and statistical analysis was performed using the SPSS software (IBM, SPSS Statistics, US, v. 23). Student's t-test was evaluated after Levine test on variance homogeneity. Where variance significantly differed, the Bonferroni approach was used in place of Student's t-test. For discrete and dichotomic variables, risk analysis was performed. Time-related tests (Kaplan-Meier survival curve; Cox regression) for survival analysis was performed using univariable and multivariable analyses. Conservative, non-parametric tests (e.g., Mann-Whitney U test) were used to evaluate the cytoplasmic and nuclear FMRP staining in cases (patients with metastases).

## Results

### Differential FMRP expression in breast tumor with metastasis (Cases) and without metastasis (Control)

Previous findings have suggested an association of FMRP overexpression with BC progression and the metastatic spread, in particular to the lungs [29]. Based on these data, we performed an immunohistochemical analysis on 127 BC tissues to obtain clinical information on the occurrence of distal metastasis. Specifically, 43 cases developed distant metastasis and 84 controls did not. Cases and controls had a follow-up of 7 years (mean) (Table 1).

As previously described on an independent cohort [29], we observed a strong FMRP staining in all the BC tissues analyzed compared to the surrounding "non tumor area", consisting of stromal cells and connective tissues, as an internal negative control of specific reaction (Fig 1). FMRP expression was higher in the controls compared to the cases and was detected in

**Table 1. Baseline characteristics in cases and controls.** Features of primary tumors of cases and controls. Number (%), Mean (± standard deviation) of tumor size and expression of receptors. The basic status of the estrogen receptor (ER), progesterone receptor (PR), cell proliferation marker Ki-67 (ki67), and human epidermal growth factor receptor 2 (HER2) play a crucial role in molecular subtypes classification of breast cancer. Student's t-test showed significant difference for all the observed variables between cases and controls, excluding size. Fisher's exact test did not show significant difference for HER2+. CI: confidence interval.

|  | Cases | Controls | p-value Student's t-test | Odds Ratio (OR) |
|---|---|---|---|---|
| **Number** | **43** *(33.9%)* | **84** *(66.1%)* |  |  |
| **Mean Size** | 2.24 (± 2.03) | 2.07 (± 1.81) | *0.555* |  |
| **ER** | 56.51 (± 38.98) | 71.71 (± 33.26) | 0.032 |  |
| **PR** | 30.18 (± 33.45) | 44.46 (± 37.82) | 0.032 |  |
| **Ki-67** | 30.35 (± 19.8) | 22.27 (± 17.99) | 0.028 |  |
| **HER2 positive** | 12 *(48%)* | 13 *(52%)* | *Fisher's exact test p = 0.095* | 2.114 (95% CI = 0.867–5.153) |
| **HER2 negative** | 31 *(30.4%)* | 71 *(69.6%)* |  |  |

both the nucleus and cytoplasm of cancer cells (Fig 1). The nuclear FMRP expression indicated a significantly lower expression of FMRP in the cases compared to the controls: 25.6 (± 27.3) vs. 43 (± 28.6), respectively, p = 0.001 (Student's t-test). The cytoplasmic expression highlighted an association of low expression of FMRP with risk of metastasis and metastatic spread, expressed by the odds ratio (OR) of 8.4 (95% confidence interval (CI): 2.8–25.4, p < 0.001) (Fig 2). We next compared the cytoplasmic and nuclear expression with a correlation test analyzing case-by-case the 127 patients. A strong correlation between nuclear and cytoplasmic FMRP staining was observed (Spearman Rho = 0.785, p < 0.001).

## FMRP expression correlates with specific subtypes

We next categorized the tumor samples according to the subtypes, and assessed the association of FMRP levels with patient outcome (Chi-squared test). HER2-positive and basal-like subtypes showed a clear difference compared to the other subtypes. In the aggressive HER2-positive cancer subtype, FMRP expression levels were lower than in the controls and lower FMRP levels were associated with a higher risk of distant relapse (Exact Fisher test, p = 0.000; OR = 7.357; 95% CI = 3.2–16.7). In the basal-like subtype, FMRP was highly expressed in the cytoplasm in both cases and controls (no statistically significant difference). This data further strengthening that FMRP levels are higher in the most aggressive breast cancers, as previously reported [29].

FMRP nuclear staining was significantly different between basal and other molecular subtypes (FMRP nuclear staining for basal-like was 58.2, mean for all subtypes was 37.1; Student t-test, p = 0.001). This analysis suggests that in the very aggressive subtype (basal-like), FMRP expression levels are already high and therefore a larger cohort would be required for a correlation analysis with distant metastasis.

## FMRP expression does not correlate with lymph-node status

Local lymph nodes are a major prognostic factor of BC. As shown in Table 2, the presence of positive axillary lymph nodes is strongly associated with the event of metastasis. Indeed, the chi-squared test for the outcome showed significant difference (p = 0.004) with an OR of 3.6 (95% CI: 1.5–8.8) (note: axillary lymph node dissection (ALND) data do not present for all). However, no correlation was found between cytoplasmic and nuclear FMRP staining, which is lower in cases than controls, and ALND, probably in line with previous findings [29] and further suggesting that FMRP is involved in aggressive breast cancer that spread via the non-lymphatic system.

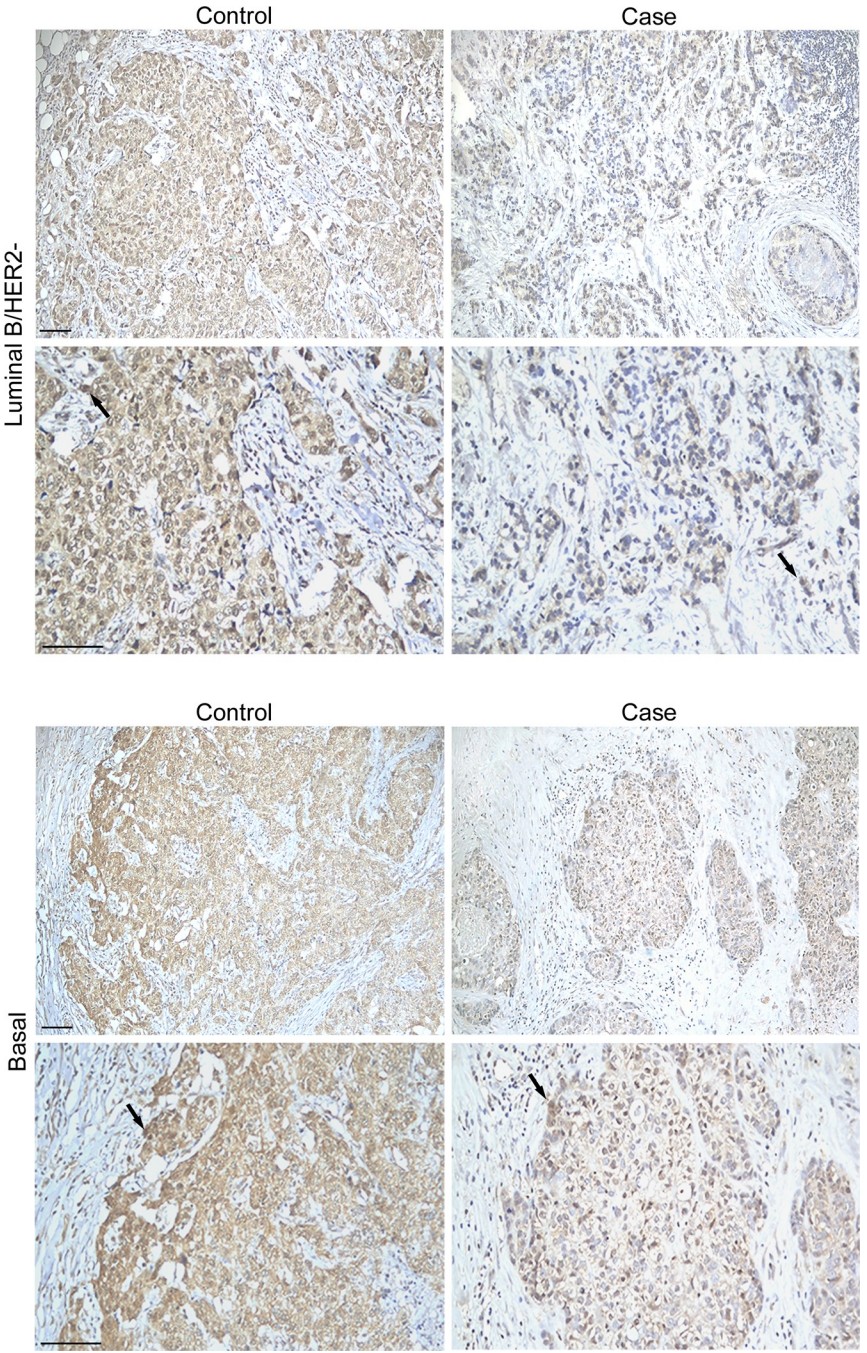

**Fig 1. FMRP expression in breast cancer patients in controls (without metastases) and cases (with metastases).** Representative microphotographs of α-FMRP immunostaining performed on primary breast cancer tissue sections from Controls (no metastases) and Cases (metastases). No metastatic tissue was analyzed in our study. Images of two different molecular subtypes, luminal B/HER2- and basal (triple negative), are included for illustration. As shown, FMRP is highly expressed in Controls compared to Cases, both in the nuclei (arrows) and in the cytoplasm of tumor cells. As described in the text, cytoplasmic and nuclear expression are strongly correlated (p < 0.001). Arrows indicate nuclear expression. Surrounding stromal cells and connective tissue are negative for FMRP staining. Scale bar = 100 μm.

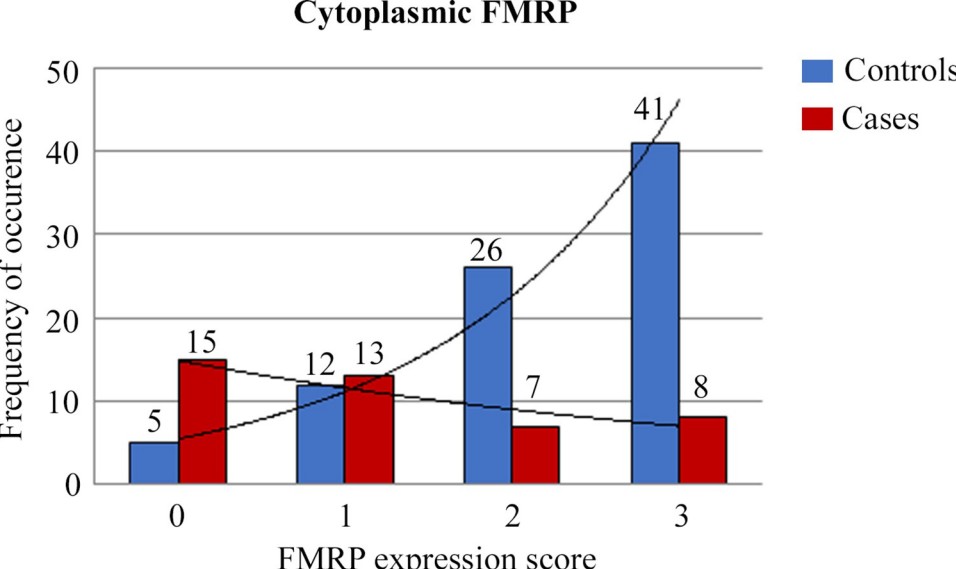

**Fig 2. FMRP expression in the cytoplasm of cancer cells in controls and cases.** The histogram represents FMRP expression in the cytoplasm from Controls in blue and Cases in red. The frequency of occurrence (number of patients characterized by a defined FMRP expression score) was plotted versus FMRP expression. FMRP levels were determined by IHC using a grading system in arbitrary units (absence of positivity (0), weak (1), moderate (2), and strong (3)). The expression of FMRP in the cytoplasm shows an inverse correlation in Cases versus Controls (Fisher's exact test, p < 0.001).

## FMRP expression correlates with the occurrence of metastasis over time

Survival tests were performed considering the appearance of metastasis (and not death) as a study event during the study (mean 7 years). We followed the association between absence, weak, moderate or high expression of cytoplasmic FMRP and the occurrence of metastasis over time (Fig 3). Each step of the curve represents an event (metastasis), each dash the end of an observation time (censored). Kaplan-Meier univariate survival analysis showed that patients with no FMRP cytoplasmic expression had more frequent metastasis (p < 0.001). The average time of observation was higher for the controls, who have not developed metastasis. The survival graph with censored events and patients with no (blue) or with (green) FMRP expression is shown (Fig 3 left panel). Similar results were obtained for the multivariate survival Kaplan Meier test (Fig 3 right panel).

Cox regression bivariate analysis recognized the lowest FMRP cytoplasmic expression as a risk (beta exponent) for metastatic outcome of 3.5 (95% CI: 1.85–6.65) and p < 0.001 (Fig 4). Also in this case, we followed the association between the absence, low and high expression of cytoplasmic FMRP and the occurrence of metastasis over time. This analysis highlighted the direct association between low cytoplasmic FMRP levels (from 3 –highest, to 0 –absence) and the risk of developing metastasis (Fig 4).

**Table 2. Axillary lymph-node (ALN) positivity.** Axillary Lymph-Node Dissection (ALND) was performed during surgery from most of the patients analyzed. ALN positivity is among the most powerful prognostic factors is of great value as an independent predictor of distant disease development. In the table, data divided into cases and controls significantly confirm that having positive ALNs increases the risk of developing a metastasis.

| | Cases | Controls | Statistics | *p-value* | Odds Ratio | 95% CI | | notes |
|---|---|---|---|---|---|---|---|---|
| | | | | | | Inf | Sup | |
| **ALND+n (%)** | 24 *(43.6)* | 31 *(56.4)* | Chi-sq | *0.004* | 3.613 | 1.475 | 8.848 | data not available for all cases |
| **ALND-n (%)** | 9 *(17.6)* | 42 *(82.4)* | | | | | | |

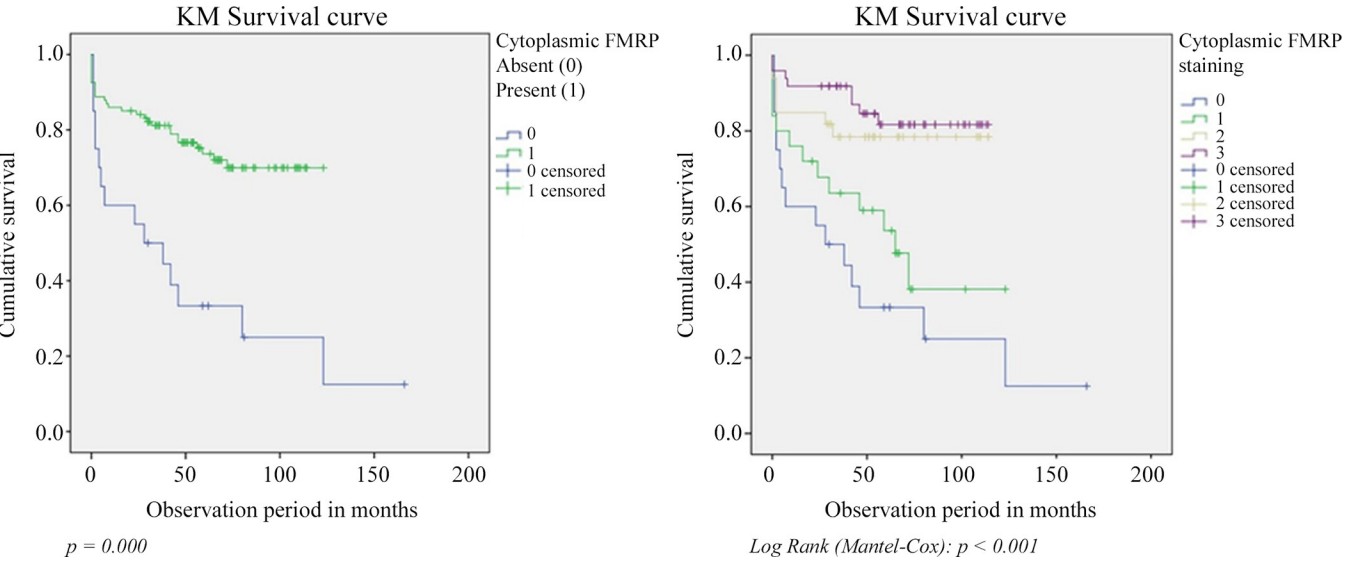

**Fig 3. Kaplan-Meier univariate and multivariate survival curves based on FMRP cytoplasmic staining.** As survival rate we considered the time of metastasis appearance (not death of the patient) during the entire length of the study (mean 7 years). In the curve, each step is an event (metastasis), each dash the end of an observation time (censored). We followed the association between absence (0), low or high (1) expression of cytoplasmic FMRP and the occurrence of metastasis over time. Left panel, Kaplan-Meier univariate survival analysis shows the curve for the absence (blue) and presence (green) of FMRP expression. Right panel, Kaplan Meier multivariate survival analysis shows the same results. Kaplan-Meier univariate and multivariate survival analysis showed that the absence or lower FMRP cytoplasmic expression had more frequent metastasis (p < 0.001).

## FMRP expression correlates with site-specific metastasis

Next, we analyzed FMRP expression in tumor tissues in relation to the frequency and site of metastasis, this part of the study involved only patients with metastases (Cases). Distant recurrence of BC at the specific site of metastasis were evaluated. Table 3 summarizes the presence of recurrences in our sample (data not available for all cases, we have three missing data for the study about the site of metastasis).

Although lower expression of FMRP was associated with increased frequency of metastasis, within metastatic cases, we observed a higher FMRP expression (nuclear or cytoplasmic) in patients with multiple metastases: the Mann-Whitney U test for independent samples showed a significant difference (p = 0.018) between the two groups (mean of nuclear FMRP on the "multiple metastasis" group = 32.07 vs. "single metastasis" = 18.84) about nuclear expression of FMRP, whereas the difference between the two groups for cytoplasmic FMRP was slightly above the threshold (p = 0.054). Second, nuclear level of FMRP was significantly higher in primary tumors that developed metastases at liver (p = 0.005; Mann-Whitney's U-test) and lung (p = 0.011; Mann-Whitney's U-test) sites and lower in patients who developed brain (p = 0.033; Student's t-test) and bone (p = 0.038; Student's t-test) metastasis (Table 4). No significant differences were observed for distant lymph node metastases. Cytoplasmic FMRP expression revealed the same trend. Cytoplasmic staining was significantly higher in primary tumors that developed metastases at liver (p = 0.018; Mann-Whitney test) and lung (p = 0.015; Mann-Whitney test) sites, and not significantly different for brain, bone, and distal lymph nodes.

## Discussion

FMRP is a ubiquitous protein highly expressed in brain where it plays a key role in regulating different processes involved in brain development and neuronal plasticity [20]. It has been

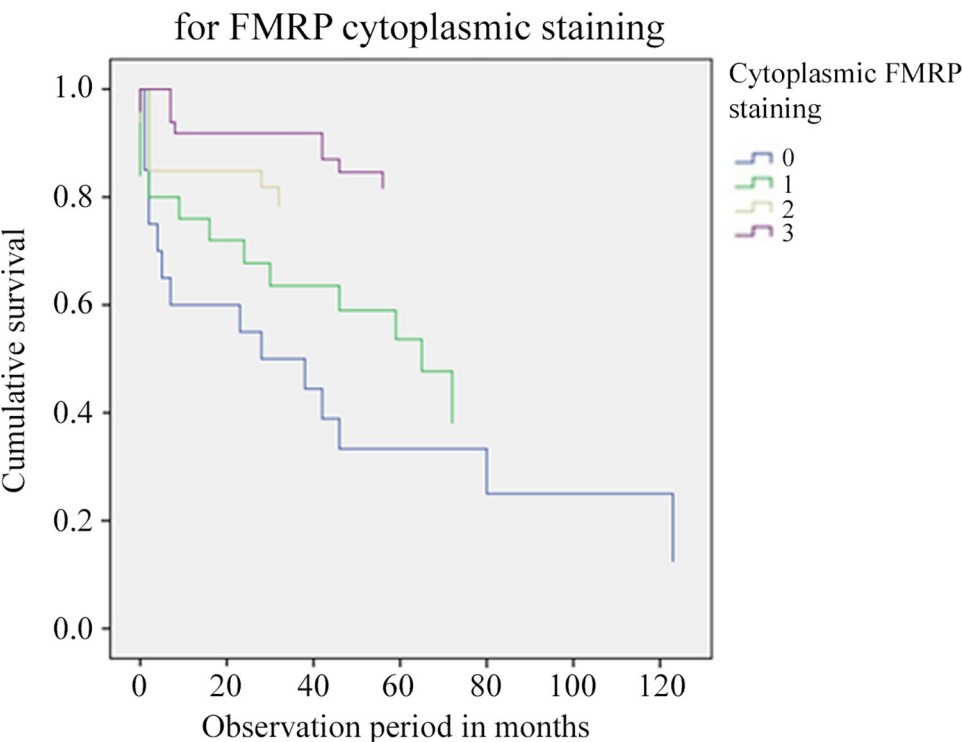

**Fig 4. Metastasis occurrence stratified for FMRP cytoplasmic expression categories.** Survival tests were performed considering the appearance of metastasis (and not death) during the study (mean 7 years). We followed the association between the scoring of cytoplasmic FMRP expression (absence of positivity (0), weak (1), moderate (2), and strong (3)) and the occurrence of metastasis over time. We have observed an increasing association between cytoplasmic FMRP levels (from 3 = highest, to 0 = absence) and the risk of developing metastasis. Cox regression bivariate analysis recognized class 0 cytoplasmic expression of FMRP as the highest risk for metastatic outcome (beta exponent as OR = 3.5; 95% CI: 1.85–6.65 and p < 0.001).

shown that lack of expression of this protein significantly reduce the risk of cancer incidence [29, 36]. Furthermore, additional reports described an association between FMRP and the development and/or severity of different cancer types, including BC where FMRP is expressed in a non-homogeneous way in the different molecular subtypes, with a greater presence in those with poorer prognosis (basal-like) [29, 37]. In the studies related to FMRP and cancer, the protein expression has been associated with the dysregulation of cellular processes such as

**Table 3. Specific sites of metastasis for 40 cases analyzed in this study.** The table summarizes the presence of recurrences in our sample. Data on the site of metastasis sites are missing from three patients. A higher nuclear FMRP expression was detected in patients with multiple metastases (Mann-Whitney U test, p = 0.018 between the "multiple metastasis" group and "single metastasis" group), whereas the difference between the two groups for cytoplasmic FMRP was slightly above the threshold (p = 0.054).

| Site | Number (%) |
|---|---|
| Multiple | 24/40 (60%) |
| Bone | 24/40 (60%) |
| Lung | 15/40 (37.5%) |
| Liver | 13/40 (32.5%) |
| Brain | 4/40 (10%) |
| Distant nodes | 13/40 (32.5%) |

**Table 4. Primary tumor FMRP expression correlated with metastatic sites.** Nuclear and cytoplasmic staining of FMRP of the primary tumor in relation to the development of metastases at different sites. "Yes" and "no" mean that the metastasis is or is not present in that organ. We used conservative, non-parametric tests (e.g., Mann-Whitney U test) for the results of both cytoplasmic and nuclear FMRP expression in cases (n = 40). Nuclear FMRP staining was significantly higher in liver (p = 0.005; Mann-Whitney U test) and lung (p = 0.011; Mann-Whitney U test) and lower in brain (p = 0.033; Student's t-test) and bone (p = 0.038; Student's t-test) metastasis. No significant differences (NS) were observed for distant lymph node metastases. Cytoplasmic FMRP staining was significantly higher in the liver (p = 0.018; Mann-Whitney U test) and lung (p = 0.015; Mann-Whitney U test), and not significantly different for brain, bone, and distal lymph nodes. Data not available for all cases (patients with metastases).

| Metastasis site | | N. | FMRP staining | | | | | |
|---|---|---|---|---|---|---|---|---|
| | | | nuclear | *p-value* | *Statistical test* | cytoplasmic | *p-value* | *Statistical test* |
| **Brain** | yes | 4 | 9.56 | *0.033* | *Student's t-test* | 0.5 | *NS* | *Mann-Whitney U test* |
| | no | 36 | 28.69 | | | 1.3 | | |
| **Bone** | yes | 24 | 21.79 | *0.038* | *Student's t-test* | 1.08 | *NS* | *Mann-Whitney U test* |
| | no | 16 | 34.25 | | | 1.43 | | |
| **Liver** | yes | 13 | 39.16 | *0.005* | *Mann-Whitney U test* | 1.84 | *0.018* | *Mann-Whitney U test* |
| | no | 27 | 20.81 | | | 0.92 | | |
| **Lung** | yes | 15 | 42.45 | *0.011* | *Mann-Whitney U test* | 1.8 | *0.015* | *Mann-Whitney U test* |
| | no | 25 | 17.37 | | | 0.88 | | |
| **Distant nodes** | yes | 13 | 27.82 | *NS* | *Student's t-test* | 1.23 | *NS* | *Mann-Whitney U test* |
| | no | 27 | 26.27 | | | 1.22 | | |

proliferation, invasiveness and EMT, which are the first step involved in the development of metastases, the major cause of cancer deaths [24, 26, 29].

Based on this evidence, our study aimed at describing the correlation between the FMRP expression in samples derived from patients with BC and the promotion/site of distant metastases. Two populations of women with BC were compared: those who developed metastases (Cases) and those who did not (Controls), during a comparable overlapping observational period, no healthy controls were used. We considered the development of metastasis as the primary outcome. For each patient, we evaluated the expression of FMRP in early primary tumor samples, revealing as expected a cytoplasmic expression and, for the first time in BC, detecting as well a nuclear staining. Each sample was analyzed according to the outcome, considering the molecular subtypes, positivity of the axillary lymph nodes, survival tests, presence of metastases, and site of metastasis.

We confirmed an increase in FMRP levels in the breast tumor compared to the apparently healthy area surrounding the tumor, consistent with previous findings [29, 37]. In addition, our analysis revealed that in basal-like tumor subtype, the worst BC associated with the shortest survival time and poor clinical outcome, the high levels of FMRP are independent from the metastatic event. Although the number of samples is low, these data suggest that FMRP is mainly involved in the aggressiveness of this tumor type rather than the metastasis development. In contrast, within the HER2-positive cases, which was the most representative subtype in our cohort, we observed that lower levels of FMRP in the primary tumor were associated with the presence of metastasis.

Growing evidence demonstrated that the tumor microenvironment, characterized from different type of cells, such as fibroblasts, endothelial cells, immune cells, etc. and a specific extracellular matrix (ECM) composition, has a crucial role in primary tumor development and metastasis formation [38, 39]. A worse patient outcome in a metastatic breast cancer is also defined by the presence of specific extracellular matrix proteins and enzymes involved in ECM degradation [40, 41]. Because several studies reported an alteration of the connective tissues in FXS, highlighting the role of FMRP in maintaining ECM homeostasis [42], it is tempting to hypothesize that FMRP could regulate the composition of tumor microenvironments, affecting the final outcome of metastasis. In addition, FMRP regulates mRNAs encoding for the

extracellular matrix proteins, such as integrins and cadherins [26, 29, 43] as well as matrix metalloproteinases (MMPs), which are involved in the degradation of the ECM [44].

Metastasis formation requires several steps including invasion of surrounding tissues, intravasation, survival in the blood circulation, extravasation and colonization/growth in the secondary organ, processes characterized by cellular and molecular peculiar changes [45]. FMRP is known to bind and regulate, positively or negatively, the metabolism of several mRNAs that encode for enzymes implicated in extracellular matrix remodeling [46–48] therefore FMRP could have a specificity of action in the different organs.

Consistently, the correlation between FMRP expression and the site of metastasis in the metastatic patients (43 patients) seems to support our hypothesis. High FMRP levels correlate with lung and liver metastases, while low FMRP levels are associated with brain and bone metastasis. Significant difference was not observed for distant lymphatic metastases. Although further interdisciplinary research is needed to define the role of FMRP, these results suggest that FMRP may have a prognostic and predictive value particularly for the metastatic behavior of the tumor.

In conclusion, our study reinforces the role of FMRP in BC, particularly in the association with the frequency and distribution of distant metastases in relation to the specific subtype. The present work further contributes to the working hypothesis that FMRP can influence the risk for developing tumor, and as such individuals with FXS might be protected from cancer. At present, the evidence is based on few epidemiological studies and case report [29, 36, 49], but it highlights the importance of further investigating the link between FMRP and cancer-related genes.

## Supporting information

**S1 Data.**
(XLSX)

## Author Contributions

**Conceptualization:** E. Caredda, P. Orsaria, O. C. Buonomo, A. Orlandi, C. Bagni, L. Palombi.

**Data curation:** E. Caredda, F. D'Amico, M. G. Scioli, P. Orsaria, G. Vanni, O. C. Buonomo, L. Palombi.

**Formal analysis:** E. Caredda, F. D'Amico, M. G. Scioli, P. Orsaria, O. C. Buonomo.

**Funding acquisition:** A. Orlandi, C. Bagni, L. Palombi.

**Investigation:** E. Caredda, G. Pedini, F. D'Amico, M. G. Scioli, L. Pacini.

**Supervision:** L. Palombi.

**Writing – original draft:** E. Caredda, L. Palombi.

**Writing – review & editing:** E. Caredda, G. Pedini, L. Pacini, A. Orlandi, C. Bagni.

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
