## [Decision Letter · Decision Letter 0]

27 Dec 2022

PONE-D-22-28908FMRP expression in primary breast tumor cells correlates with recurrence and specific site of metastasisPLOS ONE

Dear Dr. Palombi,

Thank you for submitting your manuscript to PLOS ONE. After careful consideration, we feel that it has merit but does not fully meet PLOS ONE’s publication criteria as it currently stands. Therefore, we invite you to submit a revised version of the manuscript that addresses the points raised during the review process.

We look forward to receiving your revised manuscript.

Kind regards,

James A.L. Brown, PhD, MA(Ed)

Academic Editor

PLOS ONE

Journal Requirements:

2. You indicated that you had ethical approval for your study. Please clarify whether minors (participants under the age of 18 years) were included in this study. If yes, in your Methods section, please ensure you have also stated whether you obtained consent from parents or guardians of the minors included in the study or whether the research ethics committee or IRB specifically waived the need for their consent

“This work was supported by: Telethon GGP20137, PRIN 201789LFKB MIUR to CB.”

“This work was supported by Telethon GGP20137 and PRIN 201789LFKB MIUR, to CB. The funders had no role in study design, data collection and analysis, decision to publish, or preparation of the manuscript.

Other Authors received no specific funding for this work.”

Additional Editor Comments :

Please pay particular attention to fully address the comments from Reviewers 1 and 2, which I believe will help significantly improve the clarity and impact of that paper.

Reviewers' comments:

Reviewer's Responses to Questions

**Comments to the Author**

1. Is the manuscript technically sound, and do the data support the conclusions?

Reviewer #1: Partly

Reviewer #2: Yes

Reviewer #3: Partly

2. Has the statistical analysis been performed appropriately and rigorously? 

Reviewer #1: Yes

Reviewer #2: Yes

Reviewer #3: Yes

3. Have the authors made all data underlying the findings in their manuscript fully available?

Reviewer #1: Yes

Reviewer #2: Yes

Reviewer #3: Yes

4. Is the manuscript presented in an intelligible fashion and written in standard English?

Reviewer #1: No

Reviewer #2: No

Reviewer #3: Yes

5. Review Comments to the Author

Reviewer #1: PONE-D-22-28908

FMRP expression in primary breast tumor cells…

PLOS ONE

The authors have described a potentially useful biomarker for the propensity of metastatic spread of breast cancer (BC), based on the levels of nuclear and cytoplasmic FMRP determined through immunocytochemical staining. When further developed, this approach would add to the evaluation of BC for the propensity of metastatic spread, and even the expectation for lymphatic vs non-lymphatic spread to other tissues.

However, there are several concerns with the manuscript as written that temper enthusiasm for the work.

(i) The description of FMRP measurements is confusing. Much of the manuscript appears to be describing FMRP in cancer tissue within the primary tumor in breast tissue; however, it is not clear whether additional measures of FMRP are also made in the metastatic foci. For example, in Table 4, the title is “FMRP expression in metastatic site”. This is not clarified in either the Methods section, nor in the legend to the figure.

(ii) Also with respect to Figure 4, under the header, “metastatic site”; there is a “yes” and “no” which are not defined, for each target organ, but which presumably means that a metastasis is/is not located in that organ. Also, for those cases where there is a “no”, presumably meaning no metastasis at that site – what then is the meaning of the nuclear and cytoplasmic FMRP staining, since the table refers to FMRP expression “in metastasis site”?

(iii) In Figure 2, there is an inverse correlation of FMRP expression in cases vs controls, but elsewhere in the manuscript, it is pointed out that among cases, high FMRP levels correlate with lung and liver metastases, while tissues associated with brain and bone metastasis showed a low FMRP expression. Thus, the red bars may represent the average of two very different behaviors, depending on the site of metastasis.

(iv) From the broader perspective, little attention is given to why FMRP levels are up in some types of BC, and why there may be differential expression in nuclei vs cytoplasm. Because all of the reported work is performed in situ, what steps were undertaken to demonstrate that observed differences are not due to differences in antibody accessibility following fixation from one case to the next?

(v) P16 /l334 – if absence of FMRP is associated with lower rates of cancer, why is metastatic spread associated with lower FMRP? p17 /l339 for Basal subtype – the opposite was found.

This work is not specifically concerned with the mechanistic basis for the various levels of FMRP expression; nonetheless, there should have been some discussion as to why such striking differences exist in patterns of FMRP expression.

Reviewer #2: Caredda E et al.’s manuscript titled: “FMRP expression in primary breast tumor cells correlates with recurrence and specific site of metastasis” analyzed the expression levels of FMRP in the primary and its associated metastatic lesions. Form this data, it was concluded that there was a direct and inverse correlation with FMRP expression and metastasis to the different sites.

Major Concerns:

1) There is a lack of mechanistic and functional data as to why FMRP levels (high and low expression) promote metastasis to different organs. Are there are indication of the role of the organ environment that regulates FMRP levels?

2) Is this differential expression of FMRP also seen in preclinical model of breast cancer metastasis?

3) Needs significant more experiments to suggest that FMRP levels could be used as a prognostic factor for site-specific metastasis.

Additional Concerns:

1) With respect to the latter, at the end of line 50 on page 3, given that 685,000 deaths are ~30% of the 2,261,419 cases of BC, this reviewer was uncertain how the authors arrived at a value of 6.8%. The WHO website, that the author’s provide, indicates that the total cases of all cancers is 10 million, which clarifies the issue since, of course, 685,000 is ~ 6.8% of all cancer deaths. Thus, it would be helpful if the author’s included this information; e.g., adding (red-type) to the sentence: “In 2020, the World Health Organization (WHO) reported that, of the 10 million cancers cases, 2,261,419 were new cases of BC….”.

2) Additionally, a misconception needs to be corrected in line 71 as “a single”, within the sentence’s context, is generally incorrect, given that several reports have provided evidence that clumps of tumor cells or tumor cells within groups of other cells (e.g., macrophages or fibroblasts), as often or more often have been found to seed metastasis relative to single cells. Thus, “single cells” should be deleted, which then means that “leaves” (a few words later) becomes singular.

3) At the end of line 88 there is some confusion as to whether the authors meant to refer to metastasis or tumor progression or tumor aggressiveness or a combination of these rather than tumorigenesis, given that the examples cited in the paragraph don’t refer to tumorigenesis. The sentence (lines 95 – 96) is disconnected from the paragraph that it is associated with as what is stated has not been linked to FMRP expression – please clarify this.

4) In line 328 “primitive” is inappropriate as it has been used in the context of embryological defects in neuroectodermal development that led to tumors. Therefore, delete “primitive” and replace with “nascent primary tumor” or “early primary tumor”. More importantly, throughout the Results section the findings that lower FMRP expression is associated with metastasis is very much ambiguous, due to the wording of several sentences. As such, contrary to the findings that FMRP is a suppressor of metastatic progression/lethality, a reader is left either with a sense that FMRP promotes metastatic progression or is confused as to when it does or doesn’t contribute to poor prognosis. This needs to be corrected; e.g., by: 1) in line 193, immediately in front of “FMRP” insert “low expression of”; 2) in line 216, immediately in front of “FMRP” insert “lower”; 3) emphasize in the legend to Table 2 (or in the text describing Table 2) that Cases have lower FMRP expression; 4) in line 266, immediately before “cytoplasmic” insert “low”; and, 5) in line 289 immediately in front of “We” insert “Although lower expression of FMRP was consistent with increased frequencies of metastasis, within cases of metastasis…”.

5) Adding to the confusion are statements in the Discussion that indicate that FMRP is an oncogene, apparently (referred to in an ambiguous manner) during tumorigenesis and/or primary tumor progression. As such, in these cases some explanations/distinctions between high levels of FMRP and its function in the onset/progression of primary tumor development and low levels of expression FMRP that are associated with poor prognosis need to be made.

6) Authors must add a discussion of the limitations of the small cohort size especially with respect to the very few TNBC samples studied. This was acknowledged in the Results but must be further discussed in more detail in the Discussion.

Reviewer #3: The manuscript (Luca R is well written. The study design is good with stratification into cases and controls. The study while not novel with a similar study being published by Luca R et al in EMBO in 2013, ref 29 is still a good validation and proof of principal study with low levels of FMRP in cases vs control. Additionally the association of FMRP with site- specific metastases is interesting. The role of FMRP in breast cancers especially in the various molecular types is key to understanding the aggressive subtypes of breast cancers (TNBC) which was illustrated by the authors.

The major deficiency in this paper is the lack of good quality histology pictures to illustrate the results. Did the authors examine the expression of FMRP in matched primary and metastatic tumors for example lung and brain and in the advancing front of the primary tumors. Are all tumors IDC ,NST or where they other histologic types like invasive lobular carcinoma.

Please add more figures with specific reference to whether whole sections, histologic subtypes and primary versus metastases before consideration.

6. PLOS authors have the option to publish the peer review history of their article (what does this mean?). If published, this will include your full peer review and any attached files.

Reviewer #1: No

Reviewer #2: No

Reviewer #3: No

---

## [Author Response · Author response to Decision Letter 0]

15 Apr 2023

Rebuttal letter PONE-S-22-37225 – Caredda et al.

Dear Editor,

I am pleased to submit to PLOS ONE, on behalf of all authors, a revised version of the manuscript “FMRP expression in primary breast tumor cells correlates with recurrence and specific site of metastasis”, that addresses the points raised during the review process.

I also want to thank you for your interest in this research paper and your kind cooperation.

Journal Requirements:

Regarding ethical approval (item 2), we would like to clarify that no participant in our study was under 18 years of age.

We have provided the correct grant numbers for the awards received for our study in the “Funding information” section (item 3).

Regarding “data available upon request”, we have uploaded the minimal data set underlying the results of our research to show its features (point 5).

I would also like to thank the reviewers for their important contribution to this study through the development of focal points in the development of the paper.

The following corrections have been made to the review:

Concerning the possible molecular mechanism by which FMRP regulates the metastatic event, and the possible role of organ environment in breast cancer the following part was added in the discussion: “Growing evidence demonstrated that the tumor microenvironment, characterized from different type of cells, such as fibroblasts, endothelial cells, immune cells, etc. and a specific extracellular matrix (ECM) composition, has a crucial role in primary tumor development and metastasis formation (38,39). A worse patient outcome in a metastatic breast cancer is also defined by the presence of specific extracellular matrix proteins and enzymes involved in ECM degradation (40,41). Because several studies reported an alteration of the connective tissues in FXS, highlighting the role of FMRP in maintaining ECM homeostasis (42), it is tempting to hypothesize that FMRP could regulate the composition of tumor microenvironments, affecting the final outcome of metastasis. In addition, FMRP regulates mRNAs encoding for the extracellular matrix proteins, such as integrins and cadherins (26,29,43) as well as matrix metalloproteinases (MMPs), which are involved in the degradation of the ECM (44).

Metastasis formation requires several steps including invasion of surrounding tissues, intravasation, survival in the blood circulation, extravasation and colonization/growth in the secondary organ, processes characterized by cellular and molecular peculiar changes (45). FMRP is known to bind and regulate, positively or negatively, the metabolism of several mRNAs that encode for enzymes implicated in extracellular matrix remodeling (46,47,48) therefore FMRP could have a specificity of action in the different organs.”

To better clarify the FMRP expression in the basal subtype the following part of the discussion was modified: “We confirmed an increase in FMRP levels in the breast tumor compared to the apparently healthy area surrounding the tumor, consistent with previous findings (29,37). In addition, our analysis revealed that in basal-like tumor subtype, the worst BC associated with the shortest survival time and poor clinical outcome, the high levels of FMRP are independent from the metastatic event. Although the number of samples is low, these data suggest that FMRP is mainly involved in the aggressiveness of this tumor type rather than the metastasis development. In contrast, within the HER2-positive cases, which was the most representative subtype in our cohort, we observed that lower levels of FMRP in the primary tumor were associated with the presence of metastasis.” 

Additional measures of FMRP in the metastatic foci need to be clarified: Only tissue from the primary tumor was available for the study. No analysis of metastatic foci was performed. We may have misunderstood something in the paper. We did not actually analyze the metastatic foci associated with the primary lesions. We analyzed primary tissue and followed patients over time who did or did not develop metastases. Our data are follow-up data, not direct cytological analysis of metastatic tissue.

The legends for table 4 and Figure 2 have been revised and clarified for what they are showing, and we hope that their meaning is now clear.

For Table 4, we confirm that "yes" and "no" mean that metastasis is or is not present in that organ. It should also be noted that the meaning of the FMRP staining refers only to the primary site of the tumor, which may or may not have developed metastases in that organ over time (follow-up).

We confirm that Figure 2 shows an inverse correlation between Cases and Controls, which is unambiguously confirmed by the statistical tests performed. Therefore, the averages between Cases and between Controls are strongly different (statistically significant). In another part of the article, only the behavior of the Cases is described. The variability of FMRP expression within the group of Cases is related to the different sites of metastasis. However, this variability does not overlap with the values of the Controls: the averages between Cases and Controls remain statistically different in an inverse correlation. The red bar is certainly an average, expressing a different FMRP staining according to the metastatic site, but this does not cancel out the differences found with Controls. The two are not mutually exclusive.

Unfortunately, sample size is one of the stated limitations of our work, which is why we were limited in exploring broader perspectives. This did not allow for a complete representation of all molecular subtypes, especially for TNBC.

Apart from our study (and the cited references), there are no other studies correlating FMRP and metastatic BC. There are no preclinical models of this correlation. With this article, we hope to direct research towards the study of such models.

We have replaced Figure 1 with higher quality histological images to provide a more representative and consistent illustration of the results.

A technical clarification: all samples were fixed, processed and immunostained in a standardized manner using automated machines. This eliminates any possible technical issues between samples.

With regard to the additional concerns we have changed/modified the text as suggested by the reviewers.

In conclusion, as mentioned in the discussion, further interdisciplinary research is needed to define the role of FMRP as a prognostic factor for the specific site of metastasis. We hope to further understand the biological mechanism underlying these strong correlations.

Thank you very much for your attention.

Your sincerely, 

Prof Leonardo Palombi

---

## [Decision Letter · Decision Letter 1]

29 May 2023

FMRP expression in primary breast tumor cells correlates with recurrence and specific site of metastasis

PONE-D-22-28908R1

Dear Dr. Palombi,

We’re pleased to inform you that your manuscript has been judged scientifically suitable for publication and will be formally accepted for publication once it meets all outstanding technical requirements.

Kind regards,

James A.L. Brown, PhD, MA(Ed)

Academic Editor

PLOS ONE

Additional Editor Comments (optional):

Congratulations for your work, and for addressing the reviewers comments.

Please take note of the final suggestion of Reviewer 1 to alter the wording in your abstract during the proofing process.

Reviewers' comments:

Reviewer's Responses to Questions

**Comments to the Author**

1. If the authors have adequately addressed your comments raised in a previous round of review and you feel that this manuscript is now acceptable for publication, you may indicate that here to bypass the “Comments to the Author” section, enter your conflict of interest statement in the “Confidential to Editor” section, and submit your "Accept" recommendation.

Reviewer #1: (No Response)

Reviewer #2: All comments have been addressed

2. Is the manuscript technically sound, and do the data support the conclusions?

Reviewer #1: Yes

Reviewer #2: Yes

3. Has the statistical analysis been performed appropriately and rigorously? 

Reviewer #1: N/A

Reviewer #2: Yes

4. Have the authors made all data underlying the findings in their manuscript fully available?

Reviewer #1: No

Reviewer #2: Yes

5. Is the manuscript presented in an intelligible fashion and written in standard English?

Reviewer #1: Yes

Reviewer #2: Yes

6. Review Comments to the Author

Reviewer #1: The authors have substantially addressed all of the issues that I had raised in the initial review. However, I suggest that the authors make one revision to the abstract that changes "...our study aimed at understanding the correlation..." to ...our study aimed at DESCRIBING the correlation..." since they do not provide any data that represent an attempt to understand the basis of the correlations.

Reviewer #2: The authors have satisfactorily addressed all the previous queries by providing additional data and additional explanation.

7. PLOS authors have the option to publish the peer review history of their article (what does this mean?). If published, this will include your full peer review and any attached files.

Reviewer #1: No

Reviewer #2: No

---

## [Editor Report · Acceptance letter]

14 Jun 2023

PONE-D-22-28908R1 

FMRP expression in primary breast tumor cells
correlates with recurrence and specific site of metastasis 

Dear Dr. Palombi:

I'm pleased to inform you that your manuscript has been deemed suitable for publication in PLOS ONE. Congratulations! Your manuscript is now with our production department. 

Kind regards, 

on behalf of

Dr. James A.L. Brown 

Academic Editor

PLOS ONE